Acute lymphoblastic leukemia cancer diagnosis in children and adults using transforming blood fluorescence microscopy imaging

Rehman Amjad 1
http://orcid.org/0009-0005-5751-5528 Mujahid Muhammad 1
Saba Tanzila 1
Alamri Faten S. 2 fsalamri@pnu.edu.sa
Ayesha Noor 3
1 AIDA Lab, CCIS, Prince Sultan University , Riyadh , Saudi Arabia
2 Department of Mathematical Sciences, College of Science, Princess Nourah Bint Abdulrahman University , Riyadh , Saudi Arabia
3 Center of Excellence in CyberSecurity (CYBEX), Prince Sultan University , Riyadh , Saudi Arabia
Zhou Jiayan
Electronic publication date: 2025 Aug 7
Publication date: 2025
Volume: 11
Electronic Location ID: e2997
Received 2025 May 1; Accepted 2025 Jun 10
Copyright: © 2025 Rehman et al.
Copyright year: 2025
Copyright holder: Rehman et al.
License: This is an open access article distributed under the terms of the Creative Commons Attribution License, which permits unrestricted use, distribution, reproduction and adaptation in any medium and for any purpose provided that it is properly attributed. For attribution, the original author(s), title, publication source (PeerJ Computer Science) and either DOI or URL of the article must be cited.
License URL: https://creativecommons.org/licenses/by/4.0/

Keywords: Acute leukemia cancer, Medical informatics, Blood microscopy, Image processing, Healthcare

Funding: Princess Nourah bint Abdulrahman University, Riyadh, Saudi Arabia PNURSP2025R346 The Artificial Intelligence & Data Analytics Lab, CCIS, Prince Sultan University, Riyadh Saudi Arabia This research is supported by Princess Nourah bint Abdulrahman University Researchers Supporting Project number (PNURSP2025R346), Princess Nourah bint Abdulrahman University, Riyadh, Saudi Arabia. The Artificial Intelligence & Data Analytics Lab, CCIS, Prince Sultan University, Riyadh Saudi Arabia supported the Article Processing Charges (APC) of this publication. The funders had a role in study design, data collection and analysis, decision to publish, or preparation of the manuscript.

==============================
Leukemia is a highly aggressive kind of cancer that may impact the bone marrow. The most fatal type acute lymphoblastic leukemia (ALL), is characterized by the excessive growth of immature white blood cells in the bone marrow. For diagnostic purposes, hematologists and experts use a state-of-the-art microscope fitted with a high-powered magnifying lens to analyze blood and bone marrow samples. Experts attribute the rapid progress to the presence of adolescent white blood cells, not fully developed ones. A good treatment for ALL, no matter where it comes from, includes chemotherapy, medication given through a transplant. Experts have difficulties in accurately evaluating explosive cell features due to the onerous and time-consuming nature of manual diagnosis for this disease. A total of 89 individuals suspected of having ALL underwent sample collection, resulting in the acquisition of 3,256 images. The dataset is classified into four different types of cancer: early-stage, benign cells, pre-cancerous cells, and pro-cancer cells. The proposed approach employs several preprocessing and augmentation techniques to improve the results. The studies demonstrate that the technique achieved a recall rate of 100% for the pro-cell cancer subtype, an overall accuracy of 98.67% using enhanced data, and an overall accuracy of 97.87% using the original data. The experiments have shown that the proposed equipment is superior in reliability and accuracy compared to existing approaches, and it facilitates early detection in medical imaging.

Introduction

The progression of the cancer leads to the atypical clinical manifestation of acute lymphocytic leukemia (ALL), which replaces the lymphoid organs and components of the bone marrow. The primary feature of ALL is the uncontrolled proliferation of aberrant embryonic cells and their offspring. Acute lymphocytic leukemia is the most common kind of cancer in children and usually has a favorable response to appropriate therapy (Sheykhhasan, Manoochehri & Dama, 2022). Despite a more unfavorable prognosis, adults remain susceptible to developing ALL. Among adults, it is the second most common form of acute leukemia. Moreover, ALL is the predominant category. Despite a 90% treatment rate for children with ALL, it continues to be a prominent cause of mortality for both adults and children (Iacobucci & Mullighan, 2017). The four primary classifications of leukemia are acute myeloblastic leukemia (AML), chronic myeloblastic leukemia (CML), ALL, and chronic lymphocytic leukemia. ALL is the most lethal form of leukemia, accounting for 70% of all cases. Moreover, both genetic and environmental factors have a significant influence on the onset of the illness. Uncontrolled and excessive lymphocyte proliferation in the bone marrow results in ALL (Sampathila et al., 2022). Furthermore, these malignant cells have the ability to infiltrate the circulatory system and cause significant damage to several vital organs, including but not limited to the kidneys, liver, brain, and heart. This might lead to the development of more severe tumors. The World Health Organization’s International Agency for Research on Cancer reported a total of 437,033 cases of leukemia worldwide as of 2022, resulting in 303,006 fatalities (Singh et al., 2023).

Prior to classifying and promptly commencing treatment, the recognition and determination of ALL rely on the essential examination of the physical characteristics of blood samples taken from the outside regions of the body. While it is an essential first step in detecting certain hematologic diseases, such as ALL to save lives from more deaths. These aspects include the duration and effort required for the operations, the significant level of variation across distinct observers, and the desire for proficient personnel (Wang et al., 2024; Jothi et al., 2020). Flow cytometry is critical for the thorough identification, description, and categorization of all components found in bone marrow samples via the immunophenotyping process. Nevertheless, there are other drawbacks linked to this method, including its excessively high expense, adverse impact on the patient’s overall health, and the intense pain endured during the extraction of bone marrow samples. In light of the many limitations of existing diagnostics, it is crucial to devise innovative tools that may aid in attaining a more accurate diagnosis and classification of ALL (Ahmed et al., 2019).

When it comes to effectively detecting and categorizing normal cells and abnormal cells in medical imaging, convolutional neural networks (CNN) are now considered the technique most commonly used (Shafique & Tehsin, 2018). CNN require a large dataset and processing resources for training. CNN may encounter insufficient training data from the outset. Transfer learning is a precise approach that allows us to employ deep CNN models in this context while minimizing CPU overhead (Rastogi, Khanna & Singh, 2022). The concept of transfer learning relies on the successful extension of a deep learning model’s ability to distinguish between individual classes to a new dataset. This is the core notion underpinning transfer learning. We anticipate that the models that perform the best, having undergone extensive training, will adapt to a new dataset with minimal difficulty. To achieve maximum performance, it is possible that extensive data sets are not required. This is because, in a similar vein, improving the model on a particular dataset may require a smaller amount of data. Furthermore, the resurgence in popularity of deep learning has led to a careful assessment and comparison of a large number of pre-trained, publicly available deep learning models for a variety of applications (Anwar & Alam, 2020). It is possible that a cost-effective strategy for developing models that perform extraordinarily well is to find novel approaches to changing current models in order to adapt them to a variety of circumstances. The proposed research offers the following contributions. The ResNet-CNN sequential architecture provides the basis for a novel approach of classifying acute lymphoblastic leukemia. This method utilizes transfer learning, customized layers, and a limited number of rectified linear units (RLUs) to extract crucial features from blood microscopy data.

Preprocessing enhances the image quality and resolves overfitting problems that may arise using advanced augmentation techniques on pre-trained models with medical data. These techniques will provide the highest classification accuracy and the most precise identification of acute lymphoblastic leukemia with several subtypes.

Grad-CAM helps to visualize what the proposed approach focuses on while making detections. It highlights important image regions, making decisions more transparent. This is especially useful for debugging, building trust, and ensuring reliability in healthcare fields.

We conduct a comprehensive evaluation and comparative study to demonstrate the efficacy and feasibility of the proposed approach compared to the current deep learning models.

Literature

Deep learning (DL) frameworks had shown exceptional accuracy in several medical imaging fields, such as identifying different diseases from peripheral blood samples. Computer aided diagnosis (CAD) systems are gradually adopting these methodologies. Nevertheless, so far, no technique has examined the level of focus excellence in each and every image or provided an approach for selectively enhancing samples for the purpose of classification. The approach used DL and image processing methods to standardize the cell radius, validate the quality of focus, and enhance the images through adaptive sharpening. Subsequently, the system performed the classification. The proposed methodology yielded positive results when applied to a publicly available database, including all images. Multiple cutting-edge CNNs were used for the purpose of classification (Genovese et al., 2021). The data originated from a CodaLab competition aimed at identifying leukemic cells in microscope images and distinguishing them from healthy cells. The VGG-16 network and the ResNet-50 network were used, both of which are renowned deep learning networks. The assigned parameters have previously been used to train these two networks. Therefore, modified the learning parameters and weights and also used the stored weights. Two separate groups of acute leukemia could be further defined with the aid of a neural network with twelve layers. Six fundamental machine-learning approaches were also developed. Random forest (RF) had the best accuracy score of 81.72% among all machine learning (ML) methods, while multilayer perceptron had the lowest accuracy rate of 27.33%. Concluding. This research demonstrated proposed network was the most precise method for diagnosing acute leukemia. When evaluating several networks and ML techniques for diagnosing this ailment, the one that achieved the highest performance and the fastest processing time without any delay stood out as the most effective (Das & Meher, 2021).

This study presents a highly efficient deep CNN architecture to address this problem and provided somehow accurate results for blood leukemia with quicker approach. This approach suggests the use of a unique weight factor based on likelihood, which was crucial for effectively combining ResNet18 with MobilenetV2 while retaining the advantages of both methods. The effectiveness of the system was confirmed by testing it using the two blood sample datasets (Sampathila et al., 2022). This study presents two independent classification models to identify cases of acute leukemia by analyzing the microscopic blood visuals included in a dataset. A hybrid approach combining machine learning and the AlexNet model was utilized for the detection of ALL. The process consists of three main stages: pre-processing, extract features using AlexNet, and classification using various ML algorithms. After implementing specific image pre-processing techniques, the network utilizes AlexNet for extraction and classification blood samples (Surya Sashank, Jain & Venkateswaran, 2021). All images were enhanced prior to their input into the systems by employing the average and Laplacian filters. The research resulted in three distinct frameworks. The first framework combines features from the gray level co-occurrence matrix (GLCM), and the local binary pattern (LBP). For their second proposed system, they implement the ResNet-18, AlexNet, and GoogleNet models. These are capable of accurately identifying and extracting deep feature maps due to their transfer learning-based methodology. ResNet-18, AlexNet, and GoogleNet all achieved a perfect score, indicating that the models were capable of early detection of leukemia in all three datasets (Abunadi & Senan, 2022).

The examination of a stain from a blood sample with the use of a microscope, the ALL identification procedure will become more automated and less susceptible to errors caused by human intervention. The use of artificial intelligence has already resulted in a number of advantages in leukemia detection. With its ability to examine samples for diagnostic purposes in a time-efficient manner, automated peripheral blood smear (PBS) analysis in healthcare institutions demonstrates its technological innovation (Sampathila et al., 2022). The process involves scrutinizing a blood sample under a microscope. The utilization of efficient models enhanced the automation of the ALL identification process, reducing the vulnerability to mistakes due to human error. Currently, artificial intelligence provides specific benefits in the identification of leukemia. Automated PBS analysis is a technique that enables healthcare companies to showcase their technological advancement by swiftly examining samples for diagnostic reasons (Alqudah & Suen, 2022). Das, Pradhan & Meher (2021) presented a very successful and computationally efficient method for the detection of All. The addition of dense layers and dropouts to the ResNet50 improved its design and produced three additional models. The objective of Saeed et al. (2022) was to decrease the number of deaths in the early stages of the medical sector by using existing models and a fine-tuning method based on transfer learning to forecast ALL. To enhance the precision of the model, they have included augmentation steps to address the issue of imbalanced class, thereby reducing the training errors.

Ghaderzadeh et al. (2022) employed deep learning-based approach that utilizes data to accurately diagnose ALL and its subtypes. The study’s dataset comprised 3,256 samples from 89 patients with potential ALL diagnoses. By using a cost-effective method for segmenting leukemia cells, the study included all image samples and its segmented pairs. The architecture comprises DenseNet-201 module for features and a separate block for classification. The DeneNet-201 model was trained to categorize into pro-, early pre-, and pre- ALL subtypes, as well as benign and malignant categories. The proposed multi-stage deep learning architecture improves the identification of ALL and the categorization of its subtypes by doing a more thorough analysis of images. However, it is computationally demanding and necessitates domain expertise. Atteia, Alhussan & Samee (2022) used a Bayesian approach to enhance the CNN for classification of ALL with microscopic samples. CNN uses Bayesian optimization to change its network architecture and hyper-parameters based on input data. Iteratively refining hyper-parameter space for reducing an intended error rate is the goal of this approach. Train and evaluate the optimized network using a hybrid dataset that includes two publicly available datasets. By adding more information to the hybrid dataset, this augmentation method boosts performance. This technique for enhancing data enhances efficiency.

Khan Tusar et al. (2024) developed numerous deep neural network models and improved an online tele-diagnostic tool to boost diagnosis precision. Providing affordable and conveniently available healthcare is a crucial step in combating ALL. Despite several study attempts, there is currently a lack of a complete method and universal model for using deep neural network (DNN) in diagnosing acute leukemia. Keeping the original size gave EfficientNet-V2 the greatest F1-score, recall, accuracy, and precision. The EfficientNet-V1 method attained the second-highest accuracy on a 256 × 256 pixel image. The F1-score peaked with the original picture size and EfficientNet-V1. Combining EfficientNet-V1 and V2 improved ensemble model accuracy (0.885) and F1-score (0.736). The ensemble model identified all twelve cell types. A receiver operating characteristic curve area greater than 0.9 indicates excellent functionality (Park et al., 2024). Specifically, Gupta et al. (2023) discovered that radiation therapy, when coupled with several cancer therapies, might raise the chance of “secondary” leukemia. Proteins involved in DNA repair pathways undergo modifications, leading to this. This illness is already more likely to occur if there is a history of blood problems in the family, such as leukemia, or if there are certain genetic factors, such as Downs syndrome or Fanconi anemia. The research gaps associated with the published works are represented in Table 1.

Table 1 Research gap associated with published work.

Published work	Research gap	
Genovese et al. (2021)	The proposed structure lacks clarity in its explanation and utilizes a basic CNN architecture, resulting in a relatively low accuracy of around 96%. The absence of training loss, accuracy, testing loss, accuracy, ROC-AUC, and confusion matrix is important since they are the key metrics to validate the effectiveness of models.	
Das & Meher (2021)	The proposed system exhibits lower performance reliability and specifically tackles the binary classification problem.	
Surya Sashank, Jain & Venkateswaran (2021)	Applying AlexNet for feature extraction limits the model’s application to diverse datasets or medical condition photos. AlexNet might struggle to consistently extract the most significant qualities, potentially impacting its performance on unobserved data.	
Abunadi & Senan (2022)	The model experienced overfitting, and the main constraint of the proposed research was the dataset’s size. The model’s resilience across diverse datasets was not assessed.	
Sampathila et al. (2022)	They utilized a basic CNN model that performed poorly, evaluated the model with binary data, and did not apply cross-validation or ROC-AUC to determine the error rate.	
Das, Pradhan & Meher (2021)	The deep model is used for feature extractor and machine classifiers in order to categorize disease labels. This process is time demanding and leads in detection results that are insufficient.	
Saeed et al. (2022)	The authors were employed large-scale and multi-layered EfficientNet approaches that were computationally ineffective in contrast to the other models.	
Khan Tusar et al. (2024)	A significant amount of processing capacity is frequently required for prediction and training with multi-DNN models. This situation may exacerbate the difficulty of assimilation in regions with limited resources.	

Materials and Methods

The approaches used in this work include the implementation of a proposed framework, gradient-weighted class activation mapping, methods for dataset augmentation, image pretreatment for modification, and a description of the dataset. Figure 1 provides a visual representation of the intricate design of the proposed methodology. When it comes to making reliable classifications, the preprocessing stage is very critical since it eliminates any extraneous data and emphasizes the information that is most important. The preprocessing phase is simple, fast, and has a little impact on the resources available to the system. To accurately diagnose ALL, one must adhere to the detailed iterative techniques, which involve image classification, dataset analysis, and image pre-processing. The presentation of each step in the process makes it possible for the readers to comprehend the process in a short amount of time. The breakdown of the detection process into its constituent components helps to understand the technique used. The representation dissects the framework’s structure to provide a systematic explanation of the model’s operation.

Figure 1 This diagram shows the proposed early acute lymphoblastic leukemia diagnosis methodology.

Preprocessing is essential to creating reliable classifications since it removes extraneous data and emphasizes important facts. Preprocessing is simple, fast, and takes minimal system resources. Acute lymphoblastic leukemia diagnosis requires accurate iterative methods. They include image classification, dataset analysis, and image pre-processing. The depiction of each step helps readers understand the procedure quickly. Breaking the detection process into its sections helps comprehend the technique.

The proposed methodology investigates a novel method for identifying and classifying ALL to determine the severity associated with the disease. When it comes to clinical diagnosis, determining which diseases are present might be considered challenging. This technique has the potential to improve doctors’ capacity to more accurately predict their patients’ health status through the use of microscope images, which might ultimately result in an improvement in patient satisfaction. The integration process produces useful data for the system. Edge detection, normalization, and resizing are preprocessing operations we perform on data. The proposed framework evaluates a number of different subtypes of ALL using a thoroughly preprocessed dataset.

Dataset description

This work provides evidence of the development of ALL, a data collection method that is both comprehensive and accessible to the general public. We utilized the previously described dataset for training and validate the proposed methodology. The bone marrow laboratory at Taleqani Hospital, located in Tehran, Iran, created the images included in this collection. There are 3,256 PBS photographs in the collection. With meticulous attention to detail, laboratory personnel took these photographs of dyed and processed blood samples (Aria et al., 2021). They collected samples from 89 individuals suspected of having acute lymphoblastic leukemia. Hematogones make up the first group, while early, benign cells, pre-cells, and pro-cells make up the subsequent groups. It is common practice to refer to all of the three unique subtypes of malignant lymphoblasts that make up the AL group as “ALL.” They linked a Zeiss camera to a microscope with a hundred-fold magnification to capture the images. After that, the file system saved the images as JPG files. A specialist used the flow cytometry apparatus to gain a definitive assessment of the cell types and subtypes. Ultimately, deliver segmented images that undergo color thresholding in the HSV color space, resulting in their separation into distinct segments. Figure 2 displays a few of samples from the acquired dataset.

Figure 2 The samples collected from the dataset have been classified into four subtypes, including benign cell cancer, early cancer, pre-cancer, and pro-cancer images, along with their corresponding segmented masks.

Image dataset preprocessing

The most efficient and practical strategy for enhancing the efficiency and practicality of image preparation in experiments was found to be cutting, as opposed to scaling. Edges of an image, which are sudden disruptions, may contain almost as much information as the pixels themselves. Preprocessing is very important in deep learning, especially for medical imaging, to clean the data and make it suitable for further processing. The image datasets have a large number of dimensions. When we input them into the model, the model takes too much training time, so to minimise the processing time and reduce the computational cost, we first resize the data into smaller dimensions. This hinders the ability to discern the most prominent characteristics. An image pixel’s hold information that changes when its size increases. The approximation process enlarges an image to create a large-scale image. The utilised photographs have a size of 224 × 224 pixels. Processing such enormous photos is simply impractical due to the substantial CPU requirements associated with increasing the pixel count of an image (Omar Bappi et al., 2024).

Image dataset augmentation

Deep networks in deep learning need a large amount of training data if they are to properly generalize and achieve outstanding accuracy. Sometimes, nevertheless, the scale of image data is inadequate. The image augmentation techniques are shown in Algorithm 1. The system generates synthetic training data by applying various techniques such as zooming, flipping, random rotation, and contrast modification to the provided data. Improving the performance of deep learning algorithms depends on a more extensive dataset with a complex network architecture, as the efficacy of the training dataset greatly influences their performance. To accommodate the common constraints seen in medical imaging datasets, researchers use rotation, shifting, flipping, and cropping among various data augmentation methods. Researchers define cropping in terms of patches by removing extra material and using rotation angles. we also used certain frameworks employ flipping methods and shifting operations to enhance the training dataset (Koshinuma et al., 2023). Figure 3 displays the class distribution of training data with and without augmentation.

Algorithm 1 Image augmentation techniques.

1: Input: Path for images	
2: Output: Enhanced data	
3: Prob←Probability	
4: Ap←Augmentor.Pipeline(“path”)	
5: Step 1: Rotate	
6: Ap.rotate(Prob=0.6,maxleft_rotation=10,maxright_rotation=10)	
7: Step 2: Zoom	
8: Ap.zoom(Prob=0.2,min_factor=0.7,max_factor=1.0)	
9: Step 3: Flip Horizontally	
10: Ap.flip_left_right(Prob=0.4)	
11: Step 4: Flip Vertically	
12: Ap.flip_top_bottom(Prob=0.4)	
13: Step 5: Random Distortion	
14: Ap.random_distortion(Prob=0.53,grid_width=6,grid_height=6, magnitude = 6)	
15: Step 6: Randomly Cropping	
16: Ap.crop_random(Prob=0.5,area=0.9)	
17: Step 7: Random Contrast	
18: Ap.random_contrast(Prob=0.5,min_factor=0.7,max_factor=1.3)	
19: Step 8: Samples	
20: Ap.Sample(Number_of_samples)	

Figure 3 Class distribution of training data without augmentation and with augmentation.

This demonstrates a significant imbalance in the data on the left side of the plot, but the training data on the right side has been mostly balanced by augmentation approaches.

ResNet-CNN architecture

AlexNet had the distinction of being the first CNN architecture to emerge victorious in the ImageNet 2012 competition. As a result, each successful model has included more layers in a deep neural network with the goal of reducing the error rate. This approach is effective when there are a smaller number of levels, but it has difficulties with the disappearing or exploding gradient when there are a larger number of layers. The field of deep learning frequently confronts this issue. The gradient either diminishes or significantly magnifies. As the number of layers increases, there is a corresponding increase in the error rates during both training and testing. The proposed methodology for ALL diagnosis is presented in Algorithm 2. In this architecture, the use of residual blocks effectively resolved the issue of slopes either diminishing or amplifying. In the network layer hierarchy, certain levels between the link layer and the subsequent layers do not activate the skip connection. Consequently, a residual block is created. The residual network, often known as ResNet, plays a crucial role in addressing issues related to computer vision. The ImageNet dataset, comprising 1,000 distinct categories of objects, served as the training dataset for this network. This effectively demonstrated how the remaining blocks handled the provided photographs. Every block has several levels. A layer-by-layer description of the ResNet-CNN architecture as well as its parameters are shown in Table 2.

Algorithm 2 Proposed methodology for acute lymphoblastic leukemia diagnosis.

1: Input: Images and labels	
2: Output: Diagnosis	
3: Step 1: Image preprocessing using OpenCV and other libraries	
4:  imagesandlabels←Initializeandloaddata	
5: for each i in images do	
6:     image←Firstapplysomepreprocessinglikeresizeandnormalize	
7: end for	
8: Step 2: Splitting	
9: xtrain,xtest,ytrain,ytest←train_test_split(images,labels)	
10: Step 3: NumPy Arrays for data conversion	
11:  xtrain←ConvertdatatoNumPy	
12:  xtest←ConvertdatatoNumPy	
13: Step 4: Label_Encoder	
14:  ytrain←LabelEncoder()fit_transform(ytrain)	
15:  ytest←LabelEncoder()fit_transform(ytest)	
16: Step 5: Resizing	
17:  xtrain←inputsize	
18:  xtest←inputsize	
19: Step 6: Augemtation techniques	
20:  xaugmented←Alltechniques	
21: Step 7: Training	
22:  model←Designproposedmodel	
23:  model←modeltrainingxaugmented,ytrain	
24: Step 8: Testing	
25:  yResults←Modelresultsxtest	
26: Step 9: Evaluation	
27:  performance←Usingseveralperformancemetrics	
28: Step 10: Diagnosis Results	
29: if Accuracy>95% then	
30:    Output: Achieved superior diagnosis	
31: else	
32:   Output: Again set hyper-parameters and training	
33: end if	

Table 2 A layer-by-layer description of the ResNet-CNN architecture as well as its parameters.

Description of the proposed framework	Param#	
# Load ResNet Architecture	–	
Base = ResNet101(weights=‘imagenet’, include_top=False, input_shape=(224, 224, 3))	–	
# False all layers in ResNet	–	
for layer in base_model.layers:	–	
layer.trainable = False	–	
# Design sequential framework	–	
ResNet-CNN= Sequential()	–	
# Add ResNet Architecture	–	
ResNet-CNN.add(Base)	42,658,176	
# Add CNN layers	–	
ResNet-CNN.add(Conv2D(256, (3, 3), activation=‘relu’))	4,718,848	
ResNet-CNN.add(MaxPooling2D((2, 2)))	0	
ResNet-CNN.add(BatchNormalization())	1,024	
ResNet-CNN.add(Conv2D(128, (3, 3), activation=‘relu’, padding=“same”))	295,040	
ResNet-CNN.add(BatchNormalization())	512	
ResNet-CNN.add(Conv2D(128, (3, 3), activation=‘relu’, padding=“same”))	147,584	
ResNet-CNN.add(BatchNormalization())	512	
# Add custom layers	–	
ResNet-CNN.add(GlobalAveragePooling2D())	0	
ResNet-CNN.add(BatchNormalization())	512	
ResNet-CNN.add(Dense(128, activation=‘relu’))	16,512	
ResNet-CNN.add(Dropout(0.15))	0	
ResNet-CNN.add(Dense(64, activation=‘relu’))	8,256	
ResNet-CNN.add(Dense(32, activation=‘relu’))	2,080	
ResNet-CNN.add(Dropout(0.15))	0	
# Output layer	–	
ResNet-CNN.add(Dense(4, activation=‘softmax’))	132	
# Compilation	–	
ResNet-CNN.compile(optimizer=Adam(learning_rate=0.0001), loss=‘categorical_crossentropy’	–	
# Model training	–	
history = ResNet-CNN.fit(X_train, y_train, validation_data=(X_test, y_test), epochs=25, batch_size=32)	–	

Figure 4 depicts the architectural plan for the structure’s development. The dimensions of the input are 224 by 224, and it consists of three color channels: red, green, and blue. The framework included the top layer, ResNet, in the ResNet set to facilitate the creation of a sequence model. It was crucial since we relied on our own customized levels determined by the datasets. Employing ResNet and imageNet weights in conjunction simplifies the process of transfer learning. Layers such as convolutional, pooling, activation, and fully connected will be used for testing and training purposes. Subsequently, the CNN model employs three conv2D layers, one Maxpool layer, and three batch normalization layers. Next, we include several specific layers into the framework, such as a global pooling average layer, batch normalization layer, relu function dense layer, dropout layer, two more relu function dense layers, another dropout layer, and ultimately a softmax multiclass classification dense layer.

Figure 4 The Proposed architecture of the ResNet-CNN for acute leukemia diagnosis.

The convolutional layer is a crucial component of any convolutional neural network. The CONV layer consists of K-trainable filters, commonly known as “kernels,” which may have square dimensions for both width and height. Regardless of their size, these displays completely encompass the entire quantity. In RGB photographs, the depth of the data indicates the number of channels present in the image. The number of filters applied in the preceding layer determines the volume levels in subsequent layers of the network. Batch normalization is feasible because the normalization technique calculates the means and variances of each input layer. The entire training set will be perfectly suitable. Dropout is a regularization technique that aims to prevent models from overfitting by improving accuracy while potentially compromising training accuracy. Dropout layers in our training set stochastically eliminate inputs from one layer to the next in the network architecture for each mini-batch, with a Prob of p. A typical practice is to place the pooling layer behind the convolution layer in order to decrease the spatial dimension of the data. We process the components at different depths of the input volume independently. Pooling techniques securely maintain the volume depth.

Model selection

In this study, we select ResNet-CNN because of its excellent performance in image classification, especially in medical cancer classification. ResNet uses special connections that remove noise during training, which allows us to build deep networks without compromising performance. Compared to traditional CNNs or other architectures such as VGG-16 or EfficientNet, ResNet-CNN offers excellent depth, computational speed, and accuracy. Its ability to learn useful features from high-quality medical images makes it a prime fit for the early detection of ALL in blood images. We apply batch normalization layers to stabilize the training, followed by a global pool to effectively reduce the size of the feature mapping space. Dropout layers are added to prevent overfitting. In addition, its design facilitates data accessibility and automation, which is important when working with medical datasets of varying size and quality.

Deep classifiers

MobileNetV1 achieved parameter reduction by using a depth-wise convolution. To implement a system that incorporates three levels of filtering, compression, and expansion, an expansion layer was included during the second block iteration. To further enhance performance optimization, the Inverted Residual Block technique was used. Convolutional neural networks such as Mobilenet are highly suitable for mobile vision applications due to their user-friendly nature, low computational requirements, and ability to provide satisfactory outcomes. MobileNet is used in several practical applications, including geolocation, face analysis, detailed categorization, and item identification. Depth-wise convolutions are used to apply a single filter to each input channel (Mondal et al., 2021). During a typical convolution, the filters are applied separately to each input channel.

The visual geometry group (VGG) used six different neurons; Results were obtained for the VGG16 (Sriram et al., 2022) and VGG19 (Ahmed & Nayak, 2021) models. Three 3 × 3 blocks of size 7 × 7 were used, with two blocks in a row, to give a more predictable 5 × 5 pattern. A maximum pooling operation yields differences in regression coefficients. The VGG16 network is composed of two parallel and two parallel layers. In addition, VGG19 has sixteen layers and three fully connected layers. Entire VGG networks use 3–3 layers with 1 linear layer and several non-linear layers. VGG-19 shows that in order to achieve the highest possible classification accuracy, eighteen deep layers are required.

The fundamental concept behind EfficientNet’s convolutional neural network is known as “compound scaling.” This concept ultimately resolves the long-standing dilemma of how to maximize the efficiency of model dimensions, computational time, and precision. Compound scaling enables the modification of the depth, resolution, and breadth of a neural network, which are three crucial attributes of the network (Das, Pradhan & Meher, 2021). The breadth of a neural network refers to the amount of channels present in a certain layer. Increasing the breadth of the model enhances its accuracy by enabling it to capture a greater number of complex patterns and features. Conversely, the model’s decreased bulk enhances its practicality in locations with limited resources. When it comes to increasing the depth of a network, the number of network layers is of utmost importance. Despite requiring more computational resources, more intricate models have the capability to collect more intricate data pictures. Although more complex models may need fewer computational resources, their level of accuracy may be lower. Resolution scaling refers to the process of adjusting the size of an image to a new dimension. Images with a higher resolution exhibit greater levels of detail, perhaps resulting in enhanced performance. However, they need more RAM and computational capacity. Conversely, lower-quality images are more economical in terms of resources, but they may result in the loss of delicate details.

Google researchers have developed an InceptionV3 architecture specifically designed for CNN. In 2015, the InceptionV3 architecture was introduced as the successor to InceptionV1 and InceptionV2. InceptionV3 is specifically designed for image classification, offering a combination of high computational efficiency and unparalleled accuracy (Ramaneswaran et al., 2021). Image feature extraction is easy using the InceptionV3 architecture’s full suite of transform, fusion, and initialization modules. The network can capture a series of different features at different scales and resolutions by using the initialization module, including features by using filters of different sizes. In the domains of visual query answer, object detection, and image classification, the performance of InceptionV3 consistently outperforms existing advanced models. Transfer learning is a technique that improves performance on a specific task by adapting pre-trained weights to a new data set. Typically, it uses the prior trend weighting as the central model.

Results and discussion

The purpose of the proposed framework is to develop a sophisticated multiclassification model that can provide precise diagnoses within a limited timeframe. The experiments demonstrate that the proposed method greatly enhance the accuracy of predictions by effectively anticipating the occurrence of ALL in its initial phases. To evaluate the suggested models, we use a dataset and several statistics such as accuracy, precision, AUC, and F-score. We have devised innovative techniques for data augmentation to compensate for the deficiencies of the annotated training dataset and the constraints of the dataset itself. The experiments were conducted on a GPU that operates on an operating system and has a RAM capacity of 16 GB. The experiments were conducted using the Keras framework. This study demonstrates that 80% of the images are used for training using the random selection method. A portion of 20% will be allocated for experimental purposes. Table 3 represents the parameter settings for the proposed approach.

Table 3 Parameter settings for the proposed approach.

Setting parameters	Value range	
Loss	Categorical_crossentropy	
Batch_size	32	
Learning_rate	0.0001	
Epochs	25	
Activation	Softmax	
Optimizer	Adam	

Performance evaluation

Performance evaluations are crucial for assessing the efficacy and utility of deep learning classifiers. They are important in the learning process because they help programmers improve their systems and progress further. This may be difficult to choose, but choosing a fair and accurate performance metric for your work is crucial. Classification metrics evaluate the learning models’ performance in a variety of classification tasks. Their objective is to allocate each individual data point to one or more predetermined categories. An essential criterion for evaluating performance of deep model for detection tasks is its accuracy. The proportion of accurately predicted occurrences remains constant in relation to the total number of events in the dataset. Precision is defined as the ratio of correct positive estimates to the total number of positive forecasts (Rismayanti et al., 2023). The recall rate is defined as the proportion of positive predictions out of all actual positive instances. This test evaluates the classifier’s ability to accurately identify high-quality instances. A model with high accuracy demonstrates a reduced occurrence of both falsepositives and falsenegatives. One of these indicators may be more important, depending on the situation. The F1-score is a metric that maintains a balance between precision and recall by calculating the harmonic mean of the two. When one class is much more prevalent than the other, working with imbalanced data may be advantageous.

(1) Accuracy=(TRPR+FANR)(TRRP+TRNR+FAPR+FANR)

(2) Precisionscore=TRPR(TRPR+FAPR)

(3) Recallscore=TRPR(TRPR+FANR)

(4) F1score=(2×(precisionscore)×(recallscore)(precisionscore)+(recallscore)).

Acute lymphoblastic leukemia diagnosis analysis

The classification report for the proposed method for the detection of ALL and analysis is provided in Table 4. This study demonstrates the performance of four sub-types of leukemia cancer using an original public dataset of microscopy blood samples. The pro- cancer subtype exhibited outstanding performance, scoring 100% in all settings. Additionally, the early cancer subtype obtained precisionscore, recallscore, and f1score of 96%. The benign cancer earned a recall rate of 99% and a F1-score of 96%. The pre-cancer detection model attained an accuracy rate of 100% and an f1score of 98.42%. Nevertheless, the benign cancer has a low performance compared to other cancer subtypes, with a precision of just 93%. The proposed method achieved an overall accuracy of 97.85% and a macro average f1score of 98.08%.

Table 4 Performance report of proposed approach using original data.

Cancer	Precisionscore	Recallscore	F1score	
Benign	0.9346	0.9901	0.9615	
Early	0.9645	0.9645	0.9645	
Pre	1.0000	0.9689	0.9842	
Pro	1.0000	1.0000	1.0000	
Accuracy	–	–	0.9785	
Macro Avg.	0.9748	0.9808	0.9776	
Weighted Avg.	0.9791	0.9785	0.9786	

Table 5 presents the categorization report for the proposed method to acute lymphoblastic leukemia detection and analysis. This research uses an improved dataset of microscopy blood samples to illustrate the characteristics of four subtypes of leukemia malignancy. The pro-cancer subtype performed very well, achieving recallscore of 100%, precisionscores of 99.68%, and f1score of 99.84%. Furthermore, the precisionscore, recallscore, and f1score for the early cancer subtype were 97.40%, 98.16%, and 97.78%, respectively. The benign cancer received f1score of 97.82% and a recallrate of 97.03%. With an f1score of 99.46% and an precisionscore of 99.19%, the pre-cancer detection model achieved success. The proposed technique performed well, with a macro average recallscore of 98.73% and an overall accuracy of 98.67%.

Table 5 Performance report of proposed approach using enhanced data.

Cancer	Precisionscore	Recallscore	F1score	
Benign	0.9863	0.9703	0.9782	
Early	0.9740	0.9816	0.9778	
Pre	0.9919	0.9973	0.9946	
Pro	0.9968	1.0000	0.9984	
Accuracy	–	–	0.9867	
Macro Avg.	0.9872	0.9873	0.9872	
Weighted Avg.	0.9867	0.9867	0.9867	

Table 6 calculates the results and variability of the last epochs. The highest level of precision is 98.77, achieved during epoch 9. Over the last 10 epochs, the average accuracy has ranged from 98.77 to 0.405 standard deviations.

Table 6 Last epochs results of the model.

Last epochs	Accuracy	Precision	Recall	
1	97.55	97.55	97.55	
2	98.21	98.21	98.21	
3	98.60	98.60	98.56	
4	98.35	98.39	98.28	
5	98.14	98.14	98.14	
6	97.72	97.72	97.72	
7	98.21	98.28	98.11	
8	97.97	97.97	97.97	
9	98.74	98.77	98.74	
10	98.67	98.70	98.67	
Average results	98.20	98.23	98.20	
STD	±0.3896	±0.405	±0.389	

Result analysis using accuracy and loss

Evaluating the accuracy learning_curves and loss_curves of the proposed approach are crucial in order to validate the efficacy of the detection findings in imaging field and healthcare. Figure 5 represents the epoch wise accuracy and loss during training and validation for the proposed technique. The loss, whether during training or validation, provide some sort of indicator of performance affects and provides information on the error rate. Figure 5A demonstrates that the proposed approach using the original data has worse performance compared to the augmented dataset shown in Fig. 5B. Figure 5C displays the loss while using the original dataset, whereas Fig. 5D exhibits the loss when using an augmented dataset of blood microscopy. During the first epochs, the validation loss exhibited a high value, but subsequently decreased and eventually dropped below 0.07.

Figure 5 Epoch-wise accuracy visualization as well as loss during training and validation for the proposed technique.

The receiver operating characteristic (ROC) curve typically displays TPR on Y_axis and FPR on the X_axis. The top left_corner of Fig. 6 defines the best location with a TPR of one and a FPR of zero. Consequently, having a bigger AUC is sometimes more desirable, despite the fact that it is not practicable to do so. Because the optimal strategy involves maximizing the TPR and decreasing the FPR, the steepness of ROC curves becomes an essential factor. Because the TPR and the FPR are so well-defined and easy to understand, ROC curves are often used in classification. Figure 6A displays ROC-AUC using the original data, whereas Fig. 6B shows the ROC-AUC using the enhanced data. The performance was boosted by using additional data.

Figure 6 ROC-AUC of the proposed technique with and without augmentation.

Precision-recall curves of the proposed technique with and without augmentation is shown in Fig. 7. The precision and recall scores may vary depending on the criteria. The threshold is the lowest prob required for classifying a prediction as positive. Enhancing the thresholds often leads to enhanced precision, but it often causes a decrease in the model’s recall.

Figure 7 Precision-recall curves of the proposed technique with and without augmentation.

Cumulative gain is a numerical metric used to ensure the efficiency of a model in the domains of machine learning and information retrieval, particularly in tasks involving ranking is shown in Fig. 8. This approach simplifies the evaluation of the model’s ability to rank things accurately by combining the gains (or relevance scores) of items from several metrics.

Figure 8 Cumulative gain of the proposed technique with and without augmentation.

A confusion matrix, also known as an error matrix, is a specific tabular arrangement that provides a visual representation of the outcomes of an algorithm, often used in supervised learning (unsupervised learning uses a distinct term, such as matching matrices). Results of normalized-confusion matrix with and without augmentation is shown in Fig. 9. The matrix contains actual classes in its rows and predicted classes in its columns.

Figure 9 Results of normalized-confusion matrix with and without augmentation.

Grad-CAM

Grad-CAM is a visual tool that presents choices made using a specified framework in a clear and concise manner. To predict classes in advance, it generates course localization maps that emphasize significant sections of the input image. Grad-CAM significantly streamlines the task of seeing and comprehending the understanding process of deep transfer learning models. Grad-CAM is a technique that identifies the specific areas in an image that impact the model’s prediction. This helps to improve the understanding, clarity, and reliability of the model’s decision-making process. This method offers more flexibility and precision compared to earlier procedures. Fortunately, despite its intricate nature, the outcome is unambiguous. To construct a Grad-CAM heat map, we generate a model by segmenting a high-level picture at the desired layer in order to construct a Grad-CAM heat map. We provide the entire set of interconnected layers in order to make predictions. Subsequently, we collect the output_layer, implement the loss_function, and input it into the model. We derive the gradient of our proposed model layer with regard to the model loss. Next, to superimpose the heat map onto the original images, we remove the areas that support the gradient, decrease their size, resize them, and adjust their scale. The Grad-CAM are visualized in Figs. 10A and 10B.

Figure 10 Grad-CAM visualization using the proposed technique.

(A) Grad-CAM for first blood smear image with the proposed technique. (B) Grad-CAM for second blood smear image with the proposed technique.

Analysis of transfer learning pretrained classifiers is crucial in classification and detection. Deep classifiers apply a method known as transfer learning to solve one problem using an already-trained model for another. By applying what it has learned in one task to another, the classifier may improve its generalizability and overall performance via a process known as transfer learning. The data gathered may be used to detect cancer while training a classifier to predict whether an image contains tumors or not. Generally, a neural network should dedicate its first few layers to edge detection, the middle layer to shape recognition, and the final layer to task-specific feature extraction. The process of transfer learning entails making use of the first and intermediate layers while retraining just the subsequent ones. It helps to make the most of the labeled data from the first training session. The retraining process refines the models. However, in order to retrain using transfer learning, one must isolate and concentrate on certain layers. Table 7 displays the experiments on deep classifiers.

Table 7 Analysis of transfer learning pretrained models.

Deep classifiers	Accuracy	Precision	Recall	F1-score	
InceptionV3	94.32	94.76	95.02	94.88	
VGG16	93.03	92.87	93.05	92.96	
DenseNet121	95.29	94.93	95.12	95.02	
VGG19	92.12	92.18	92.28	92.23	
MobileNetV2	92.88	93.19	93.86	93.52	
EfficientNet-B7	96.07	96.73	96.18	96.45	

Comparison with existing studies

Deep transfer learning and fusion approaches provide a significant improvement compared to their previous versions. Previously, classical machine learning techniques like random forest and manually chosen feature sets were used. However, their use was limited by the non-uniform nature of anatomic-pathological data sets and their reliance on domain expertise. In addition, many algorithms need several pre-processing and feature extraction phases, which may be a lengthy procedure. Moreover, mistakes might arise due to human intervention. Comparison with existing studies is shown in Table 8. Genovese et al. (2021) used 260 blood samples from ALL databases and obtained an accuracy of 96.8% using VGG-16. However, the attained accuracy is considered to be somewhat poor. Das & Meher (2021), also obtained a very low level of accuracy while using the random forest methodology. Khan Tusar et al. (2024) used DNN and obtained a 97% accuracy rate. Bhuvaneswari et al. (2023) used the VGGNet model and obtained a remarkable accuracy of 95.7%. Najjar et al. (2023) obtained a 96.1% accuracy by using the hue saturation value method. Keerthivasan & Saranya (2023) used a modified CNN to diagnose cancer and obtained a remarkable accuracy of 97.6% by using all 3256 samples. Dangore et al. (2024) used CNN and attained a remarkable accuracy of 96.1%. Visual representation of comparison with existing studies is shown in Fig. 11. The previous research used same datasets and samples to diagnose ALL cancer, although yielded less precise results. The proposed strategy produced superior accuracy by using the proposed preprocessing and sophisticated augmentation approaches, as deep learning demonstrates enhanced performance with larger datasets.

Table 8 Comparison with other studies.

Authors	Datasets	Approach	Accuracy	
Genovese et al. (2021)	ALLIDB2, 260 samples	VGG-16	96.84	
Das & Meher (2021)	ALLISBI 2019	Random forest	81.72	
Khan Tusar et al. (2024)	ALL Bone Marrow	DNN	97.00	
Bhuvaneswari et al. (2023)	ALL Bone Marrow	VGGNet	95.75	
Najjar et al. (2023)	ALL Bone Marrow	Hue saturation value	96.13	
Keerthivasan & Saranya (2023)	ALL Bone Marrow	Modified DCNN	97.69	
Dangore et al. (2024)	ALL Bone Marrow	CNN	96.16	
[Proposed]	ALL Bone Marrow	ResNet-CNN	98.67	

Figure 11 Visual representation of comparison with existing studies (Genovese et al., 2021; Das & Meher, 2021; Khan Tusar et al., 2024; Bhuvaneswari et al., 2023; Najjar et al., 2023; Keerthivasan & Saranya, 2023; Dangore et al., 2024).

Statistical T-test

We used a statistical t-test to measure the difference in the performance of the proposed model using augmented and without-augmented data as illustrated in Table 9. In all cases, we utilised the last five epochs’ accuracy, precision, and recall of the proposed model with and without augmentation and applied a t-test. We obtained t-test values of 3.97004, 3.43585, and 3.46484, along with p-values of 0.01654, 0.02639, and 0.02570 for accuracy, precision, and recall, respectively. The test indicates that if the p-value is <0.05, there is a significant difference, while if the p-value is >0.05, then there is no difference. In our case, our values indicate that there is a significant difference in the performance when comparing with the augmented data and improvement in the accuracy. Also, in case 1, the first row indicates the mean score + STD without augmentation, and the 2nd row indicates with augmentation. Also, using the mean score, we obtained 0.0062 STD without augmentation and 0.0039 STD with augmentation, ensuring that with augmentation, the proposed approach achieved a minimum STD score.

Table 9 Statistical T-test and standard deviation.

Comparison	T-statistic	P value	Mean score+STD	
Case 1	3.97004	0.01654	0.9724 ± 0.0062	
			0.9826 ± 0.0039	
Case 2	3.43585	0.02639	0.9733 ± 0.0062	
			0.9829 ± 0.0041	
Case 3	3.46484	0.02570	0.9721 ± 0.0062	
			0.9824 ± 0.0040	

Discussion

The proposed ResNet-CNN architecture is used in the study to demonstrate ALL diagnoses utilising blood fluorescence microscopy imaging. A variety of augmentation approaches are employed to improve the dataset, and both original and augmented data are used in the studies. For the benign cancer class, the suggested model obtained 93.46% accuracy and 96.15% F1-score; for the early cancer class, it obtained 96.45% precision, recall, and F1-score. The model’s precision for the early cancer class was 97.40% and for benign conditions it was 98.63% when using enhanced data. VGG19 had the lowest accuracy among the pretrained models, whereas EfficientNet had the highest. The Grad-CAM facilitates the visualisation of the areas that the suggested method concentrates on during detections. It makes judgements more transparent by highlighting significant areas of the image. This is particularly helpful in the healthcare industry for debugging, establishing credibility, and guaranteeing dependability.

Artificial intelligence has a real impact on healthcare. It is not just about achieving predictive results; practical implications, such as usability, ease of use, and predictability in clinical practices, are also crucial. To achieve this, we employed a lightweight ResNet-CNN sequential architecture that requires approximately 73 seconds per training epoch for augmented data, utilising a standard CoLab notebook and GPU hardware while minimising the use of computing resources. This approach could be especially useful for hospitals or diagnostic centers that do not have access to robust surgical systems. The significance of computational demands in model development, particularly in a clinical context, lies in the speed at which an individual can execute the procedure. The design of the proposed model enables remote monitoring, rendering it perfect for situations necessitating immediate diagnostic assistance. This efficiency suggests strong potential for integration into clinical workflows, particularly in settings with limited computational resources. While training is relatively rapid, real-time or near-real-time inference is even more crucial for clinical deployment. We will focus on further optimising the model for inference speeds and evaluating deployment strategies, such as edge computing or cloud-based platforms, to ensure secure and scalable integration into routine diagnostic processes. Our current dataset is anonymised and publicly available through Kaggle, and we are fully aware of the ethical requirements surrounding the use of AI in healthcare. Issues such as patient consent, data security, and compliance with regulations will need to be carefully addressed in future real-world applications.

Conclusion and future work

ALL is a blood malignity that specifically affects the blood and bone marrow. It is distinguished by the atypical and disproportionate generation of undeveloped cells. It is the predominant form of cancer in both adults and infants. If cancer has spread to other parts of the body, surgical removal may be required. Various variables impact a patient’s prognosis, including as the kind and stage of their ailment, their age, and the effectiveness of their current medicine. Through the administration of suitable medication and early provision of subsequent care, a substantial proportion of patients may attain full remission. The objective of this research was to validate and compare the ability of proposed approach with the existing work. This was done by analyzing evaluation metrics.

We perform the experiments both with and without augmentation techniques. The recommended technique involves examining and analyzing many metrics, such as effectiveness, precision_recall curves, cumulative gain, accuracy, F1-score, recall and loss, in order to assess their efficacy and reliability. According to the data, the approach demonstrated a 100% accuracy rate in identifying the pro-cell cancer subtype and a 99.01% recall rate for benign cell cancer. In addition, the proposed approach attained an overall accuracy of 97.85% without any data augmentation (using the original data). Furthermore, during the augmentation process, the approach achieved a total accuracy of 98.67%, reducing the risk of overfitting. The recommended method’s validation loss decreased, and it obtained a high proportion of accurate predictions with a reduced number of wrong ones. The recommended model outperformed the present techniques. Our future plans include expanding our study in collaboration with other cancer databases. In order to reduce computational expenses and enhance the accuracy of detection, we shall use feature selection techniques.

Supplemental Information

Supplemental Information 1 Experimental Code.

Supplemental Information 2 Implementation Steps.

Reproducibility

Additional Information and Declarations

Competing Interests

The authors declare that they have no competing interests.

Author Contributions

Amjad Rehman conceived and designed the experiments, analyzed the data, performed the computation work, prepared figures and/or tables, and approved the final draft.

Muhammad Mujahid conceived and designed the experiments, performed the experiments, prepared figures and/or tables, and approved the final draft.

Tanzila Saba conceived and designed the experiments, analyzed the data, performed the computation work, authored or reviewed drafts of the article, and approved the final draft.

Faten S. Alamri performed the experiments, analyzed the data, performed the computation work, prepared figures and/or tables, and approved the final draft.

Noor Ayesha performed the experiments, analyzed the data, performed the computation work, authored or reviewed drafts of the article, and approved the final draft.

Data Availability

The following information was supplied regarding data availability:

The dataset for the experimentation is available at Kaggle: Mehrad Aria, Mustafa Ghaderzadeh, Davood Bashash, Hassan Abolghasemi, Farkhondeh Asadi, and Azamossadat Hosseini. (2021). Acute Lymphoblastic Leukemia (ALL) image dataset [Data set]. Kaggle. https://doi.org/10.34740/KAGGLE/DSV/2175623.

The code is available in the Supplemental File.

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
