# Peer review of "Acute lymphoblastic leukemia cancer diagnosis in children and adults using transforming blood fluorescence microscopy imaging"

_PeerJ Computer Science, doi:10.7717/peerj-cs.2997_

## Round 0.1 · original submission · Minor Revisions

The manuscript presents a promising deep learning approach for automated diagnosis of ALL, achieving high accuracy and recall. However, according to the reviewers' comments, it requires attention to several issues: 1) a lack of discussion on ethical considerations like patient consent and data privacy, 2) unclear availability of code for replication, and 3) missing class distribution data that could clarify potential imbalances. Additionally, 4) there are inconsistencies in abbreviation usage, figures presented out of order, and claims of performance improvement that lack convincing evidence due to a minimal margin of increment. We would like to invite you for a resubmission after addressing these points would enhance the clarity and rigor of the study.

Reviewer 1 ·

Basic reporting

The manuscript is written in clear, professional English, with no apparent ambiguities. The language is suitable for the interdisciplinary audience of PeerJ Computer Science, ensuring accessibility to both computational and medical researchers.
The introduction effectively contextualizes ALL, a prevalent and aggressive cancer, and highlights the limitations of manual diagnosis, such as time consumption and error-proneness. It includes a well-referenced literature review, citing studies on existing methods like VGG-16 and Random Forest, which establishes the relevance and foundation of the research.
The paper adheres to a standard research article structure, including Introduction, Methodology, Results, Discussion, and Conclusion, aligning with PeerJ standards and computational biology norms.
The introduction clearly articulates the motivation: to develop an automated, accurate, and efficient system for ALL diagnosis. This is supported by discussing the clinical need for early detection and the drawbacks of current methods.
As a machine learning study, the paper presents empirical results rather than mathematical theorems. It provides detailed metrics, including accuracy (98.67% with augmented data, 97.87% with original data), precision, recall, F1-score, ROC-AUC, and loss curves. These are clearly reported, with class-wise performance for benign, early, pre-cancer, and pro-cancer categories, meeting the criterion for formal results.

Experimental design

The research is well within the scope of PeerJ Computer Science, focusing on computational methods for biological applications, specifically automated ALL diagnosis using deep learning.
The investigation demonstrates high technical rigor, with comprehensive descriptions of preprocessing, model architecture, training parameters, and evaluation. Sources are adequately cited, particularly in comparisons with prior methods. The literature is appropriately paraphrased, enhancing the paper’s academic integrity.
The methods are described with sufficient detail to enable replication. The dataset, comprising 3,256 images from 89 individuals, is publicly available on Kaggle. The manuscript details preprocessing (edge detection, normalization, resizing), data augmentation (rotation, zooming, flipping), model architecture (ResNet101 with custom layers), training parameters (Adam optimizer, learning rate 0.0001, 25 epochs, batch size 32), and computing infrastructure (64-bit OS, 32GB RAM, core i7 3.80GHz CPU). However, it does not explicitly mention whether the code or reproduction scripts are available, which would further strengthen replicability.
The paper includes a thorough discussion on data preprocessing, covering techniques like edge detection, normalization, resizing to remove noise, and augmentation to increase dataset size and improve generalization. These steps are both necessary and sufficient, given the small size of medical imaging datasets, and align with best practices in deep learning.
The evaluation methods are well-described, using standard classification metrics: accuracy, precision, recall, F1-score, ROC-AUC, and loss curves. Class-wise performance is reported, with metrics ranging from 97.40% to 99.46% across categories and 100% recall for pro-cancer cells. The use of Grad-CAM for visualizing critical image regions adds transparency, crucial for medical applications. However, model selection methods are not explicitly discussed; a brief justification for choosing ResNet-CNN over other architectures would enhance clarity.

Validity of the findings

The conclusions are well-stated and limited to the results, claiming high accuracy and superior performance compared to existing methods. These claims are supported by detailed metrics and comparative analyses.
The experiments are conducted satisfactorily, with an 80% training and 20% testing split. Multiple metrics and visualizations like Grad-CAM ensure robust evaluation. The 100% recall for pro-cancer cells is particularly significant for diagnostic reliability, though the high performance raises questions about dataset specificity. The argument is well-developed, linking the introduction’s problem statement to the proposed solution and its validation. The study meets its goal of improving ALL diagnosis accuracy and efficiency, with results directly addressing the motivation.
The proposed method represents an advancement in automated ALL diagnosis, with potential clinical benefits. The public dataset encourages meaningful replication, and the rationale—improving early detection—benefits the field by addressing a critical healthcare need. The conclusion identifies limitations, such as computational demands and the need for domain expertise, and proposes future directions, including expanding to other cancer databases and using feature selection to reduce costs and enhance accuracy. This addresses unresolved questions and outlines a clear research path.

Additional comments

The manuscript is a strong contribution to computational biology, presenting a promising deep learning approach for ALL diagnosis. Its strengths include 1) High Performance: Achieving 98.67% accuracy and 100% recall for pro-cancer cells is impressive and clinically relevant. 2) Detailed Methodology: The comprehensive description of methods, from preprocessing to evaluation, ensures transparency.
3) Public Dataset: The availability of the dataset on Kaggle promotes replicability and further research.
4) Clinical Potential: The study addresses a real-world problem, with potential to improve early ALL detection.
However, the absence of discussion on ethical issues, such as patient consent or data privacy, is a notable oversight. This should be addressed, especially given the medical context. And it does not clarify whether the code or reproduction scripts are available. Providing these would enhance replicability. The paper does not provide the class distribution (benign, early, pre-cancer, pro-cancer), which would clarify potential imbalances and the effectiveness of augmentation. Finally, a discussion on how the tool could be integrated into clinical practice, given computational demands, would enhance its practical relevance.
Overall, the paper is a valuable contribution with minor revisions needed to address these points.

Reviewer 2 ·

Basic reporting

Rehman et al proposed a new approach to improve existing deep learning models for classifying images into different subtypes of leukemia. The authors proposed several preprocessing and augmentation methods to improve accuracy and reliability compared to existing methods. The authors collected 3256 images from 89 individuals and performed training either using original data or augmented datasets. They found the augmented dataset can help reduce overfitting. The manuscript is written well, and results presented clearly. With that said I have several comments.

Experimental design

1. For abbreviation throughout the paper, the authors should first state the full term when first time using it, such as CNN, RF, etc, whenever is possible. Please also stick to the same abbreviation throughout, such ALL instead of All.
2. From figure 4-8, please put the figures in the same order as stated in the caption, “with or without augmentation”. Now looks like it’s flipped.

Validity of the findings

For the AUC result in figure 5, I am not very convinced on “the performance was boosted” given the margin of increment is very tiny.

---

## Round 0.2 · accepted · Accept

Thank you for responding to the comments. We don’t have further suggestions. The manuscript will be transferred to the journal staff and production team for publication. Please feel free to contact us if you have any questions in the later stages.

Reviewer 2 ·

Basic reporting

I have no other comments

Experimental design

I have no other comments

Validity of the findings

I have no other comments